# Mitigating Visual Ageism in Digital Media: Designing for Dynamic Diversity to Enhance Communication Rights for Senior Citizens

**Loredana Ivan** [1,*] , **Eugène Loos** [2,*] **and George Tudorie** [1,*]

1 Communication Department, National University of Political Studies and Public Administration, 012244 Bucharest, Romania

2 School of Governance Utrecht University, 3511 ZC Utrecht, The Netherlands

* Correspondence: loredana.ivan@comunicare.ro (L.I.); e.f.loos@uu.nl (E.L.); george.tudorie@comunicare.ro (G.T.)

**Abstract:** This paper advocates for the importance of visual communication rights for older people to avoid "visual ageism," described as media practices of visually underrepresenting older people or misrepresenting them in a prejudiced way. It aims to present a set of policy recommendations using "designing for dynamic diversity" as the leading principle. By discussing studies about the ways older people are visual represented in digital media content, the paper shows how visual communication rights for older people could help to fight "visual ageism." It also pleads for collaborative ways to create digital visual content "together with" older people and not "for" them. Moreover, the paper makes a plea for empowering senior citizens by advocating their right of having a voice about the manner in which they are visually represented and enhancing their power to influence specifically the images representing them.

**Keywords:** visual ageism; visual representation; communication rights; older people; senior citizens; designing for dynamic diversity; empowerment

---

## 1. Introduction

Ageism is the process of systematic stereotyping and discriminating against people because they are old [1]. Ageism is an umbrella concept [2] consisting of beliefs, attitudes expectations, behaviors shared by community members towards older people. Ageism presents itself and is experienced at the micro-level, for example in interactions between service providers and older people [3], and at the same time encompasses societal practices, recurrently verbalized and visualized in different social contexts (see [4,5] and Section 3.1). Institutional ageism is more implicit and holds an ambivalent meaning [6]. On the one hand, senior citizens are the focus of political discourse, because older people tend to be an important segment of the actual voters (for example, in 2019, 55% of the total voters for the European elections were 55 years and above, and this percentage has been relative stable for the past five years) (https://data.europa.eu/euodp/en/data/dataset/-2019-post-election-survey-first-results). On the other hand, governments act to prevent a perceived social burden of the increased older population, by making people responsible for the way they age [4,5]; namely, whether they do this "successfully" or not [7,8].

Ageism frames the way policymakers shape policy design, legitimizing paternalistic values, leaving senior citizens out of the policy conversation and ultimately affecting their quality of life [9]. Institutional ageism [6] implies forms of passivity and acceptance of dominance directed towards older people, reinforcing each other and making difficult for everybody to think outside the box.

To give an example of institutional ageism, we refer to Lloyd-Sherlock et al. [10], who present current strategies in the Global Health Policy of WHO's actions plans aiming to reduce premature mortality from cardiovascular disease, diabetes, cancers, and respiratory disease by 25% between 2010 and 2025. In this case, "premature mortality" refers to 70 years and below, thus discouraging data collection at the national level on people over 70 years of age and the reallocation of the resources from older people to younger groups. The same study revealed forms of institutional aims in The United Nations reports on HIV and sexually transmitted diseases, which excluded people 50 years and over. The institutional ageism in such cases translate not only in national policies, accentuating and justifying the already existing age discrimination in healthcare resources, but also excluded older people from many of the epidemiological studies—as a 70-year threshold applies also in the clinical trials: the idea of reducing premature mortality comes along with the preconception that survival after the age of 70 is less important than survival at a younger age. Many other examples of institutional ageism are to be found when looking at the missing data about people over 70 years in the official statistics, including the ones provided by the Eurostat data base (https://ec.europa.eu/eurostat/data/database), see also Rosales and Fernández-Ardèvol discussing structural ageism in big data approaches [11].

Our society is ageing, which has life style implications (see for example https://aging.com/guide-to-living-a-healthy-lifestyle-at-an-old-age/). Media content related to such issues becomes more and more visual, especially in the case of digitalized media. Visual media, to a large degree institutionally produced, risks to convey implicit forms of ageism (see also Section 3.1), and therefore, visual communication rights are more than ever necessary. In this paper we, therefore, focus on a specific aspect of institutional ageism: "visual ageism," a phenomenon that we define as media practices of visually underrepresenting older people or misrepresenting them in a prejudiced way [12] (see Sections 2 and 3). We pay attention to international bodies having initiated discussions on the rights of older individuals (see Section 3.7 for more information). For instance, the United Nations have established the Open-ended Working Group on Ageing (OEWG) in 2010, with the purpose to strengthen the protection of older people's rights. While the basic human rights of older individuals are emphasized, there are areas that remain unaddressed or rather ignored [13,14], such as the visual communication rights of senior citizens: the right of having a voice about the manner in which they are visually represented and the power of older people to influence specifically the images representing them is crucial [4,5]. Therefore, this paper aims to present a set of policy recommendations using "designing for dynamic diversity"as the leading principle. Moreover, the paper makes a plea for empowering senior citizens, by advocating their right of having a voice about the manner in which they are visually represented and enhancing their power to influence specifically the images representing them.

## 2. Research Outline

Empirical studies on the presence of ageism in visual digital media content are scarce, thus, we explored trends and patterns, and as this paper does not aim to synthesize the evidence for the impact of one or more variables on human behavior, we conducted not a systematic but a narrative literature review "in which the findings ( . . . ) of relevant studies are outlined and discussed with a view to presenting an argument about the conclusions that can be drawn from the current state of knowledge in a field." ([15], p. 579). We started our literature search with a recent review by Loos and Ivan [12] focusing on "visual ageism," a notion they coined for the media practice of visually underrepresenting older people or misrepresenting them in a prejudiced way. Then, we continued by discussing papers related to the relatively unexplored phenomenon of visual ageism in the digital media content, and made a plea for the communication rights of older people, by suggesting policy recommendations to enhance senior citizens' visual communication rights. Finally, we used the concept "designing for dynamic diversity," a notion coined by Gregor et al. [16] as a leading principle to fight visual ageism in the digitalized media content, showing the importance of visually representing older people as a diverse group, in a non-stigmatized way, to enhance their communication rights. The paper

includes: an overview of the digital practices to represent older people in digital media; the key findings from previous studies in which we have revealed visual ageism in the digital content of public institutions; a plea for the importance of the communication rights for senior citizens, and how such rights could be implemented when visual content is concerned, by using the principle of designing for dynamic diversity.

## 3. Findings

*3.1. Visual Ageism—Visual Practices to Represent Older People in Digital Media Content*

Ageism in the media has been particularly treated as an asymmetric power structure based on age, a constructed justification of inequalities between different age groups, legitimizing domination [17]. By systematically under- or misrepresenting older people, media content legitimizes the dominance and reinforces the logic according to which the social construction of ageing is made and maintained [18]. Media content (textual and visual) is a continuous reflection of societal practices. It influences everyday interactions, including the way we relate to older people, as well as the way we see ourselves as "being old."

Ageism in the media content has been investigated by analyzing the presence of older people in the media, by focusing on their under-representation, their portrayal often lacking positive attributes in their given minor or peripheral roles and, generally speaking, not reflecting the characteristics of the audience (see Zhang et al. [19] for a systematic literature review).

In the past decades we have seen a gradual increase in the presence of older people in the media content and a switch towards more positive representation [17]. This change happened particularly for younger older people—below 65 years of age [20]. Older people, especially the third agers (more information follows below), are more present in traditional media (television programs, print press, advertising, and movie series), represented as active and maintaining a healthy life style, while fourth agers continue to be underrepresented. The increased visibility of older people in the media is part of the dominant discourse of "ageing-well"—an implicit form of ageism in which people are held responsible for the universal and irreversible ageing process [4,5]. This "positive" trend started especially in television and printing advertising, where older people were spotted by marketing strategists as potential valuable consumers already in the early 1990s [12].

As stated in Section 1, nowadays, media content is becoming more and more visual, especially when we talk about the digitalized media. Consequently, visual media convey implicit forms of ageism: older people are presented in couples, happy and enjoying life, but not everybody has a partner or has the resources to travel, not to mention the dominance of white people in visual media [4,5] (see also Section 3.1). Furthermore, this "ageing-well" ideology is putting pressure on the individual [4,5], and it leads to the marginalization of the aging process with the exclusion of the older elderly, especially those who are no longer able to enjoy "successful ageing" [7,8]. The different ways people create meaning in their lives as they age are often ignored. We agree with Ylänne (p. 369) [20]) stating that: "what might be considered 'positive' portrayals can turn out to be more ambiguous in their construction of older age than might at first appear to be the case."

The concept of "visual ageism" that we introduced earlier to describe the media practice of visually underrepresenting older people or misrepresenting them in a prejudiced way is particularly useful when we research the way older people are represented in digital visual media content. Visual ageism describes visual representations of older people in peripheral or minor roles without positive attributes; non-realistic, exaggerated, or distorted portraits of older people; and over-homogenized characterizations of older people.

We will discuss now some empirical studies revealing the visual ageism in the digital content of public institutions. Loos [4,5,21], for example, gained insight into the ways senior citizens' organizations in the Netherlands portrayed their members on their websites. The older people were without exception shown as enjoying an active lifestyle in the 'third age' [22,23], a long period of being well, for example

by engaging in sports or taking a leisurely time. The 'fourth age,' painful descent into decay [24], was absent from the digital content of the three Dutch websites from senior citizens' organizations included in the research [4,5] that showed that older people with a non-white ethnicity background were a minority, that older people gained visibility as being healthy and vital, in the company of others, and that the range of images excluded frail people and people from different ethnic backgrounds.

Similar empirical studies have been conducted in other countries and drew similar conclusions regarding the stigmatized ways older people are visually represented: "a dominant visual representation of older people as happy, socially involved and extroverted, while representations of older people as weak, introverted and alone constitute a minority" (pictures' analysis from DaneAge association's website, 2016–2018, Danish advocacy group for older people (p. 111 [25]). In a study conducted by Xu [26] in Sweden using picture analysis from documents of Swedish municipality guidelines, as well as an in-depth visual analysis of Facebook photos published by the municipality in 2018, the conclusions were similar: the visual portrayal of citizens is communicated using a set of traits stereotypical attributed to different life stages. Specifically, these findings suggest that the visual content "serve to categorize older people as a vulnerable group, while perpetuating age stereotypes and ageist perceptions in society" [26], p. 93. In conclusion, senior citizens' organizations, and other public institutions (such as municipalities) visually promote a positive image of older people, while at the same time representing them as excluded from other age groups and from culture and society in general.

A shift towards a more diverse visual representation of older people on the digital content of public institutions has been recorded recently. Thus, an empirical study conducted by Sourbati and Loos [3], p. 275, in the UK analyzed pictures of the people social care services homepages run by local authorities in the five largest most ethnically diverse cities where older people comprise a growing diversity of ethnic groups (according to national Census data): Birmingham, Bradford, Leicester, Manchester, and the City of London. This study identified three patterns in visual imagery: "(1) stereotypical representations of group membership as homogenous in terms of age groups, sex, health status and ethnicity, with older people typically represented as white, (un)healthy men or women; (2) new visibilities, of older people as socially and culturally diverse groups; and (3) new approaches to inclusive digital service design where age becomes a demographic variable." Moreover, the findings that resulted from a cross-country study analyzing pictures of senior citizens' organizations websites in Finland, Italy, the Netherlands, Poland, Romania, Spain, and the UK, conducted by Loos et al. [27] between December 2016 and February 2017 showed that senior citizens' organizations tend to represent older people together with others and not alone, a form of counter discourse to the dominant "ageing-well" ideology.

In recent years, we acknowledge a trend of visually representing older people using younger elderly people. This is not only an unrealistic picture, but it also puts a lot of pressure on older persons, as such a picture communicates that a younger looking body and face are of higher value than an older looking body and face. It reinforces the idea of the older age as something that people need to conceal their features, and obtain a "younger look." However, one cannot be a teenager forever. Furthermore, the picture sends a more subtle message, because it implies that the failure to adhere to this stereotype of looking younger is the responsibility of the individual. Katz and Calasanti [7] warned that "what might be considered "positive" attributes in the depiction of old age could in fact be a normative construction which has nothing to do with the real experience of older people in everyday life." Thus, even "positive" images could be harmful, creating obstacles related to the way we think about the natural ageing process: they could pressure us to conceal our wrinkles and being constantly preoccupied to look young and healthy. At the society level, policymakers could use age thresholds following the same logic: the overrated idea of "active aging" underpin many of the policy papers addressing older people [28].

### 3.2. Communication Rights for Senior Citizens

The second stanza of Yeats' Sailing to Byzantium [29], p. 2311 has had less of a public life than the haunting first line of the poem. "An aged man is but a paltry thing,/A tattered coat upon a stick, unless/Soul clap its hands and sing, and louder sing." The intended reading was spiritual restoration, which in itself is revealing. Stereotypes about older people dictate that "only in art and the domain of the spirit are they licensed to continue to be creative" [30], p. 28. The still precarious position of ever growing ageing populations find themselves in involves, among other things, a social station that often approximates invisibility and voicelessness. This is not simply a matter of being shown or talked about, but one of having a say about the manner in which one is presented, of exercising control on the images and discourse one is supposed to recognize oneself in. Yeats' "paltry thing" motive is, after all, but one iteration in a long line of pervasive negative stereotypes that have dominated visual and literary culture, and that continue to inform some current representations.

The prescriptive force of such representations should not be underestimated, even if they may now have metamorphosed into seemingly benign avatars, e.g., images of "successful ageing" [7,8]. Restrictive social roles are still at play; heterogeneity is still ignored. Adhering to such representations continues to be part of "knowing how to be old," i.e., they do what they have always done: put older people in their places. Looking back at about three decades of research on ageing and stereotypes, Palmore [31] argued that despite increased awareness, older people continue to internalize negative beliefs about their age (being "weak, sick, or senile"), and Levy [32] has extensively researched the way internalized stereotypes negatively impact older people's well-being and their quality of life. Gullette [33], p. 181, even suggests a viral negativity: "Old age" is so unsayable it needs a euphemism; "ageing" was and still is used in its place, so "ageing" too has come implicitly to signify decline."

The sheer magnitude of this problem—its history, legacy, and the number of people affected—invites a discussion in the framework of rights. Modern societies have expanded their envelope of moral concern or regard to include discriminated and vulnerable individuals, and marginally even non-human [34]. Perhaps the strongest manifestation of such concern is raising the question of rights to correct ethical blind-spots and (re)affirm the dignity of the rights-bearers. In the context of rapidly ageing societies, especially in the global North, the UN has initiated discussions on the rights of older individuals. For example, a 2011 Report of the UN Secretary General dedicates a section to "Participation in policymaking, political and cultural life" (UN Secretary General) [35]. However, the focus of such efforts is generally on urgent matters, such as poverty and health care—or on the potential economic contribution of the older people. As the basic human rights of older individuals are emphasized, there are specific areas of concern, such as those sketched above, that should also be addressed. We suggest, specifically, connecting the discussion of how older individuals are represented to the larger debate about communication rights [36–43].

Moreover, a focus on visual representations and thus on putative communication rights in this sphere is timely, given the position such representations have come to occupy in the current socio-cultural landscape. Interventions guided by principles of social justice, such as those typically embodied in a framework of rights, could help to mitigate "visual ageism."

A right to communicate has been proposed initially in 1969 in connection to the right to be informed, which is codified in the Universal Declaration of Human Rights (https://www.un.org/en/universal-declaration-human-rights/). The driving concern as this debate was emerging was the problematic power gap between citizens and media organizations and other major institutional actors. Passivity does not mix well with democratic citizenship, and transnational media organizations tend to dominate the communication landscape. This concern continues to be relevant, as corporate media consolidation continues, and as giant new/social media companies pose new challenges to active and informed citizenship [43].

In the early 1980s, the UNESCO MacBride Report explained that, while the goal of democratization of communication was consensual, the content of a right to communicate was still to be determined. The commission suggested a generous mix of association, information, and development rights that

extended from discussion and inquiry to culture and privacy [44], p. 173. It is best to keep an inclusive framework of this kind in mind as we turn to the problems faced by older individuals. Analogies are available in this context, and they are helpful in determining an adequate manner of discussing communication rights for older persons. Other vulnerable groups can be taken as proxy. The ability to consume, share, and generate information is a fundamental resource for both public participation and personal autonomy. Some isolated communities are also information-deprived, and, as such, individuals in these contexts may have less space for action and choice [45]. One could characterize their situation as an incomplete exercise of rights otherwise nominally recognized (e.g., equal worth and dignity). Similar issues are faced by some older individuals, including in the sense of not being in an equal position, relative to other persons, to share or determine the contours of information about themselves.

Children and people with communicative disabilities also make for informative comparisons in this respect, and indeed the question of the communication rights of these populations has been raised [46]. Older persons have often been infantilized (the venerable 'second childhood' motive), and while they may not literally lack a voice, their ruminations, ideas, or statements tend to be relegated to the unimportant or irrelevant. If in previous social arrangements older individuals acted as repositories of knowledge, in modern societies this role has been eroded. What older individuals have to say is often interpreted as expressing their orientation toward the past, or their narrow preoccupation with a needy present [30], pp. 28–32. Finally, while older individuals are increasingly numerous, their status could be compared to that of linguistic minorities. In fact, the two categories intersect. The difficulties of older persons who have to acquire a second language later in life, for example their struggling with their self-conception, are instructive [46]. As communication is increasingly mediated by technology, older persons have to learn new skills, including a new vocabulary that is important for their social inclusion. The right to participate tacitly becomes a duty or obligation to acquire the dominant lore of the day, which is not without irony, since media literacy itself has been discussed in terms of communication rights [47].

Precedents and analogies should help, but there is also a sense in which what we confront is unprecedented. Affluent societies have rapidly growing populations of older individuals—vulnerable constituencies at a time of exploding inequalities. Information and Communication Technology is now an essential infrastructure in most societies, which puts disproportionate pressure on the older persons. Moreover, communication practices are increasingly dependent on visual content, a process that does not happen in a cultural void, but in a context already infused with problematic images of ageing. Put together, these tendencies may create a "perfect storm"—a substantial communication divide that may confine older persons to a position of enduring inferiority.

Defining and enforcing a series of rights in this domain could be conceived of as both reparation and prevention. Damages already produced by ageism in terms of relegating senior citizens to stereotypical peripheries of communication could be thus curtailed, and the risk of further abuse mitigated. It is in any case a matter of social justice that older individuals have a say in how they are portrayed. With the framework of communication rights as horizon, communities could in the meantime target specific deleterious effects of negative stereotypes about senior citizens. Especially in the case of stereotypical visual content (see also Section 3.1), which rapidly takes a life of its own in (digital) media, there is an unmistakable urgency to do so.

*3.3. Designing for Dynamic Diversity: Policy Recommendations to Enhance Senior Citizens' Visual Communication Rights*

We agree with Ross et al. (p. 3) [48] that images can injure. "Pictures are highly emotional objects that have long-lasting staying power in within the deepest regions of our brain. But both textual and visual media messages that stereotype individuals by their concentrations, frequencies, and omissions, become part of our long-term memory." We can discuss at least three ways in which ageist visual representations are harmful:

(1) They not only reflect societal practices, but also produce potentially negative and reinforce these practices [49].

(2) They are internalized and directly influence older people's lives. "When individuals reach old age, the ageing stereotypes internalized in childhood, and then reinforced for decades, become self-stereotype" [50], p. 204. Implicit self-stereotyping improves memory in old age, increases longevity, and has longitudinal benefit on the functional health of older persons [32].

(3) Visual ageism enhances the risk for older people to have difficulties to identify or relate with pictures in which they are represented [4,5]. This is a challenge to our civil society, as it does hinder the access to information as a 'primary good' [38–40]. The equal right of access to information should be considered a basic right of all citizens, comparable to the classic (human) rights [42].

Knowing that older people are visually represented in a homogeneous and stigmatized way, we could ask ourselves which role pictures play for the identification processes of a varied group of senior citizens and the implications for accessible digital information retrieval. A fundamental question is if those who cannot identify with the way older persons are represented in pictures turn away from the organizations that present the related digital information and, as a consequence, do not use it. Loos examined this question by selecting five sets of stock photos that were used on the Web site of ANBO, a Dutch senior citizens' organization, representing a variety of older people (age, sex, living arrangements, vitality) reflecting their current life situation and accompany information about pensions, income, health, and housing, during the spring of 2018 [21]. Then, older people from different age groups (50–59, 60–69, 70–79, and 80+), living conditions (living alone, living together with others), and with different levels of vitality (self-reported) were interviewed to examine the extent to which they identified with different kinds of stock photos. The study showed that people tend to identify with stock photos that relate to their life stage and current situation.

We, therefore, make a plea for using "designing for dynamic diversity," a notion coined by Gregor et al. [16]: "making accessible interfaces for older people is a unique but many-faceted challenge" [16], p. 155. In our opinion, "designing for dynamic diversity" can also be used as a leading principle to allow organizations to avoid the pitfall of visual ageism and to design pictures for the digital media content (such as websites) to visually represent senior citizens as a diverse group, in a non-stigmatized way. "Designing for a dynamic diversity" is important, as older people are often represented as the homogeneous other, even though the differences between people increase when they get older, a phenomenon called 'aged heterogeneity' [51,52]. It can help organizations to fight visual ageism by increasing the chance that senior citizens, for which they deliver goods and services, can identify with the pictures that represent them, and make a step forward to guarantee their communication rights.

In the next section we will present a set of policy recommendations derived from the research studies on visual ageism in the media presented above, with the aim of enhancing and preserve the communication rights for older people. The recommendations are not prioritized, nor are they comprehensive, but they indicate the need of proper regulations on visual media content with the potential to harm older people.

### 3.4. Use the Principles of Designing for a Dynamic Diversity

The concept of "Designing for a Dynamic Diversity" introduced by Gregor et al. [16] could underpin current visual practices of digital content by avoiding the pitfall of representing older people as a homogeneous group in a stigmatized way. We need to develop a set of recommendations to stimulate public and private organizations to consider this concept when they visually represent older persons on their websites or social network platforms. Particularly, we can raise awareness by the new media platforms (such as for example Facebook, Instagram, and Twitter) of the potential harmful content of the stereotypical representations of older people and the importance of following the recommendations, with benefits for their older users. Additionally, we should promote education of people involving in designing and creating digital media content for different public and private organizations, whose jobs

will be to pay attention to recommendations and avoid visual ageism. Subsequently, it is important to educate people who are part of the decision process in creating and distributing digital media content, and also students to raise awareness on visual ageism and communication rights for older people. Some of the university curricula have introduced courses on communication rights, in the attempt of educating students for the new Information and Communication Technology era—in which inequalities and social exclusion could be created also by the differences in ICT access and use, and media representations of different social groups. We believe that a broad extension of such courses while including the topic of visual communication rights will be an efficient way to fight ageism in the digital media.

### 3.5. Design Not "for," but "with" Older People

There are already numerous voices advocating for the value of collaborative ways in designing products, prototypes, and media content with an older audience in mind [53,54]. Such voices argue for the value of designing "with" older people and not necessarily "for" a stereotypically defined older population and including older people in all stages of the process of designing and creating prototypes: from the general idea to the marketization and distribution of the final "product." We agree with Cutler [55] that one facet of the ageist portrait of the older population has to do with people's lower willingness and capability to learn and with their decreased openness to change. Many of the ageist views are held by young people, resulting in a bias about the development and designs of different prototypes.

Digital media content is designed mostly by young people with the youth market in mind, creating outputs that might be difficult for older people to relate with and algorithms that often fail to predict their habits, interests, and values. The use of a collaborative approach to fight visual ageism would mean to understand how older people perceive and feel about different kinds of pictures, how these pictures are combined, and how to integrate the pictures in the general communication context with the input from the audience that designers have in mind. This would be an essential step in the designers' well-informed decisions on inclusive images enhancing visual communication rights for senior citizens.

### 3.6. Avoid Offensive Digital Media Content

Generally speaking, researchers from media studies and media communications have been preoccupied with ageism in the media content. However, media studies are often criticized for the overuse of content analysis as a method, the lack of theoretical discussion [56], and the fact that they focus on the sender and neglect the receiver in the communication process [12]. Such an approach has an important limitation—that fact that what is "offensive content" is judged by specialists and not by the audience. Few studies that we know about (the study of Loos [21] discussed above is an exception) have included older people's opinions and reflections on what kind of visual messages might be offensive to them. We believe that a collaborative approach in producing and distribution visual content in the new media would benefit from a more in-depth understanding of the visual content that has a potential to harm older persons. What are the visual elements older people encounter in everyday lives and to whom they feel offended, uncomfortable or injured? We have evidence, for example, that older people are using self-deprecating humor to fight negative stereotypes largely shared in many societies regarding older people [57] (as for example the fact they are sexually un-active [58–60]) and it is important to research not only what is visual offensive content for older people, but the mechanisms they use to counteract visual ageism.

### 3.7. Visual Communication Rights for Senior Citizens

Finally, we make a plea for considering visual communication rights and their importance for various social groups—particularly older people. The right to be (re)presented in a non-prejudiced way has been constantly supported by important international organizations, as for example *Age Platform*

*Europe* (a platform that advocates for the rights for older people in EU) and the United Nations. Recently, the UN Department of Economic and Social Affairs [61] has organized an event on confronting ageism and empowering older people. The event stressed how different forms of ageism contribute to growing inequality and advocated for policy frameworks addressing older people's risk for exclusion as a result of ageism. A position paper resulted including policy recommendations. Though the importance of communication rights for older people is not underlined as such, one key policy recommendation indicates the need to "adapt learning opportunities, content and methodologies to the interests and preferences of older persons" [62]. We believe that such programmatic papers should be more specific on fighting ageism in media and protecting the communication rights of older persons.

In the United States, *AARP Foundation*, one of the most powerful advocacy organization focused on older Americans, has just released a report analyzing media images of older people showing perpetual visual ageism. For example, less the 5% of images portrayed older people with technology and the few advertising campaigns that targeted older consumers represented them as selfish and out of touch. As a result, *AARP Foundation* [63] pressed the advertising agencies to use more realistic portrayals of older people and coordinate with one of the biggest stock media supplier, Getty Image, to introduce images of older people presenting them in less stereotypical instances.

We agree with Sawchuck [64], who advocated for the necessity for seniors to use digital media as a tactic to engage politically. In fact, there are some initiatives of older people to get the attention of policymakers. A participatory action research project from 2015, Exercising Senior Citizenship in an Ageist Society [65], had as a primary objective to understand how social media platforms could be used by senior citizens for purposes of social advocacy. The project initiated an online blog (carewatch.tumblr.com) and stimulated older people to engage in conversation about different forms of ageism. The project reveals the fact that older people need proper training to use social media to strengthen their voices in their advocacy efforts, but also the fact that older people often remain frustrated in their efforts to reach policymakers on issues regarding ageism. Instead of treating older people as "victims" of the ageist media content, we need policy frameworks that will empower them in fighting offensive content. Unfortunately, projects as the one described here are merely sporadic and should be supported by policy plans of action. In this context, it is not to be neglected that the UN established an open-ended working group on ageing (OEWG) [66] in 2010, aiming at strengthening the protection of human rights of older persons in the framework of The Universal Declaration of Human Rights.

## 4. Discussion

In the current paper, we presented the importance of designing visual content "together with" older people and not "for" them. The paper makes a plea for the importance of fighting "visual ageism" and advocates for the communication rights of senior citizens.

Various forms of ageism have been researched over time, as ageism is an umbrella concept [19]. Here, we focused on "visual ageism," a term introduced before by Loos and Ivan [12], describing social practices to visual represent older people in a prejudiced way. Nowadays, media tends to be more visual, especially the new digitalized media. Consequently, approaching visual ageism is timely. We argued about three directions in which visual ageism could be analyzed in the digital media content.

First, ageism is a social practice creating asymmetric power structure and justification of inequalities between people and younger age groups. In the visual communication, such practice manifests in under- or misrepresentation of older people in stereotypical or peripheral roles. We provided evidence that such practice is spread also in the way public institutions visually represent older individuals in the digital media content.

Second, we witness a gradual increase of the presence of older people in digital media content and a more positive representation over the years. We reveal the fact that this happened mainly for the younger elderly—below 65 years of age—whereas the oldest-elderly remained rather invisible in the new media. One particular explanation lies in the fact that these younger elderly are nowadays

considered valuable consumers. Another explanation is linked with the dominant discourse of "ageing-well "—making people responsible for staying active and productive for longer periods of time. Here, we argue that "fighting ageism" will mean to include older people's diversity in digital media content and to present more realistic pictures of the older age. Representing older people always in couples, happy, travelling, and enjoying the benefits of their "well-deserved" pensions is also a form of exclusion for those who have no partners or resources to travel [4,5]. Additionally, the fact that white people dominate the digital media content, at least in the representations of older people, is an issue that needs further exploration. Generally speaking, the representations of older people tend to reinforce and reproduce some of the classic inequalities: with oldest-elderly, those with low income and/or poor health, and minorities being rather underrepresented. Such practices of visual ageism need to be addressed, and the current article does this by advocating for the visual rights for the senior citizens and proposing an inclusive approach of representing them in digital media.

Third, a new trend emerged in the recent years: representing older people as young ones and setting unrealistic standards for them—the pressure to look good at any age and appear young in body in face [4,5]. Hence, we face a different kind of visual ageism—the apparently positive images of older people, sending the subtext that a younger look is a matter of "choice," and that some individuals manage to "defeat" the natural ageing process, while others have failed. Such visual practices could link with some of the social phenomena that are insufficient researched on the older population: for example, eating disorders and depression and the preferences for aesthetic surgeries and bodily adjustments at any risk. Again, the implications of this "new visual ageism" (obsessive representations of older people in looking unrealistically young) is insufficiently approached in the literature and not addressed in the current policies advocating for communication rights. Thus, the current article opens the discussion of the visual aspects that might hurt older people and discusses the idea of designing for a dynamic diversity as a way of creating more inclusive digital content.

## 5. Conclusions

Digital media is merely visual and conveys implicit forms of ageism. Visual ageism describes visual representations of older people being in peripheral or minor roles without positive attributes; it involves non-realistic, exaggerated, or distorted portraits of older people. Such images could be harmful, creating obstacles to the way of thinking about ageing at the individual level and at a more project-oriented level for policymakers.

Damages already produced by ageism in terms of relegating senior citizens to stereotypical peripheries of communication could be curtailed, and the risk of further abuse mitigated. Thus, we make a plea for visual communication rights, having in mind the principle of "designing for dynamic diversity"—namely, designing pictures for digital media content to visually represent older people as a diverse group, in a non-stigmatized way. Additionally, designing for dynamic diversity is a collaborative process in which we co-create visual content "together with" and not "for" older people. Ultimately, older people could learn to advocate for their own rights using digital media content—in the form of online platforms and advocacy groups to create awareness about their communication rights.

Digital media is a source of implicit forms of ageism, among which visual ageism. Knowing what are the visual elements older people feel uncomfortable with or offended by when they are exposed to media content is an important step in creating a set of regulations and policy recommendations to fight visual ageism and empower senior citizens.

**Author Contributions:** Conceptualization, E.L. and G.T.; methodology, L.I.; validation, L.I., E.L. and G.T.; formal analysis, L.I.; investigation, L.I.; resources, E.L.; data curation, L.I.; writing—original draft preparation, L.I.; writing—review and editing, E.L.; supervision, E.L.; project administration, E.L. All authors have read and agreed to the published version of the manuscript.

**Funding:** There was no external funding.

**Acknowledgments:** This paper is part of the research project BConnect@Home (https://www.jp-demographic.eu/wp-content/uploads/2017/01/BCONNECT_2017_conf2018_brochure.pdf), funded by the JTP 2017—JPI More Years, Better Lives (Grant Agreement 363850)—the Netherlands, ZONMW (Project 9003037411).

**Conflicts of Interest:** The authors declare no conflict of interest.

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
