# Peer review of "Mitigating Visual Ageism in Digital Media: Designing for Dynamic Diversity to Enhance Communication Rights for Senior Citizens"

_societies, doi:10.3390/soc10040076_

Round 1

Reviewer 1 Report

 A very interesting study with good recommendations for how to address this problem. While reading this I was struck by my own blind spots. 

Author Response

REVIEW 1

Comments and Suggestions for Authors

A very interesting study with good recommendations for how to address this problem. While reading this I was struck by my own blind spots. 

  • We would like the reviewer for her/his kind words!

Reviewer 2 Report

In this manuscript, the authors deal about how visual communication rights for older people could help to fight “visual ageism” and pleads for collaborative ways to create digital visual content ”together with“ older people and not “for”  them.

Moreover, the paper makes a plea for empowering senior citizens by advocating their right of having a voice about the manner in which they are visually represented and enhancing their power to influence specifically the images representing them.

In order to increase the value of the manuscript I have the following suggestions:

  1. Introduction section: I suggest to add brief data related to a link between lifestyle and ageing.
  2. A flow chart of the study of a summarized scheme of this paper will be welcomed
  3.  I suggest adding a Discussions section in which the authors highlight the advantages, novelty and particularity of their manuscript compared to others already published.

The current article contains valuable information. The article looks very informative and would be useful for the research community. Consider revising accordingly.

Author Response

REVIEW 2

  • We would like to thank the reviewer for the valuable comments that we addressed below and in the revised manuscript.

In this manuscript, the authors deal about how visual communication rights for older people could help to fight “visual ageism” and pleads for collaborative ways to create digital visual content ”together with“ older people and not “for”  them.

Moreover, the paper makes a plea for empowering senior citizens by advocating their right of having a voice about the manner in which they are visually represented and enhancing their power to influence specifically the images representing them.

In order to increase the value of the manuscript I have the following suggestions:

  1. Introduction section: I suggest to add brief data related to a link between lifestyle and ageing.
  • Thanks for the suggestion. In the Introduction we referred to

https://aging.com/guide-to-living-a-healthy-lifestyle-at-an-old-age/

  1. A flow chart of the study of a summarized scheme of this paper will be welcomed
  • We have added the following paragraph explaining the flow of the study – in the section. We agree with the fact that the summary of the study needs to be underlined and we proceed in using a paragraph instead of a chart. We think this approach goes more in line with the rest of the paper

 A summarise scheme of the current research includes: an overview of the digital practices to represent older people in digital media; the key findings from the previous studies in which we have revealed visual ageism in the digital content of public institutions; a plea for the importance of the communication rights for senior citizens, and how such rights could be implemented when visual content is concerned, by using the principle of designing for a dynamic diversity.

  1. I suggest adding a Discussions section in which the authors highlight the advantages, novelty and particularity of their manuscript compared to others already published.

We did introduce a Discussion section

The current article contains valuable information. The article looks very informative and would be useful for the research community. Consider revising accordingly.

Reviewer 3 Report

This study is to explore mitigating visual ageism in digital media with a design for dynamic diversity to enhance communication rights for senior citizens.

The study background and objectives are provided properly. Study method and study processes are valid and sound.

Study findings and concluding remarks are proper. There are some comments to improve the quality of the paper.

Current paper has three (3) sections. Typical organization in a paper may have at least four (4) sections.

See an example at the Journal homepage. Also, add study purpose in the section 1. Section 2 may need to be extensive more. References styles are not consistent each other. Revise them. 

Author Response

REVIEW 3

Comments and Suggestions for Authors

  • We would like to thank the reviewer for valuable comments that we addressed below and in the revised manuscript.

This study is to explore mitigating visual ageism in digital media with a design for dynamic diversity to enhance communication rights for senior citizens.

The study background and objectives are provided properly. Study method and study processes are valid and sound.

Study findings and concluding remarks are proper. There are some comments to improve the quality of the paper.

Current paper has three (3) sections. Typical organization in a paper may have at least four (4) sections. See an example at the Journal homepage.

  • The paper has now 5 sections.

Also, add study purpose in the section 1.

  • We added the purpose at the end of section 1.

Section 2 may need to be extensive more.

  • We extend Section 2, by adding a paragraph and we include a Discussion section

References styles are not consistent each other. Revise them. 

  • We revised them in the manuscript.

Round 2

Reviewer 2 Report

The manuscript has been improved and now warrants publication in Societies.